# Extracellular vesicles in human semen modulate antigen-presenting cell function and decrease downstream antiviral T cell responses

**Lucia Vojtech**[1]*, **Mengying Zhang**[2], **Veronica Davé**[3,4], **Claire Levy**[1], **Sean M. Hughes**[1], **Ruofan Wang**[1], **Fernanda Calienes**[1], **Martin Prlic**[3,4,5], **Elizabeth Nance**[2,6], **Florian Hladik**[1,3,7]*

1 Department of Obstetrics and Gynecology, University of Washington, Seattle, Washington, United States of America, 2 Molecular Engineering and Sciences Institute, University of Washington, Seattle, Washington, United States of America, 3 Vaccine and Infectious Disease Division, Fred Hutchinson Cancer Research Center, Seattle, Washington, United States of America, 4 Department of Global Health, University of Washington, Seattle, Washington, United States of America, 5 Department of Immunology, University of Washington, Seattle, Washington, United States of America, 6 Department of Chemical Engineering, University of Washington, Seattle, Washington, United States of America, 7 Department of Medicine, Division of Allergy and Infectious Diseases, University of Washington, Seattle, Washington, United States of America

* luciav@uw.edu (LV); fhladik@fredhutch.org (FH)

**Data Availability Statement:** All relevant data are within the paper and its Supporting information files.

## Abstract

Human semen contains trillions of extracellular vesicles (SEV) similar in size to sexually transmitted viruses and loaded with potentially bioactive miRNAs, proteins and lipids. SEV were shown to inhibit HIV and Zika virus infectivity, but whether SEV are able also to affect subsequent immune responses is unknown. We found that SEV efficiently bound to and entered antigen-presenting cells (APC) and thus we set out to further dissect the impact of SEV on APC function and the impact on downstream T cell responses. In an APC–T cell co-culture system, SEV exposure to APC alone markedly reduced antigen-specific cytokine production, degranulation and cytotoxicity by antigen-specific memory CD8+ T cells. In contrast, inhibition of CD4+ T cell responses required both APC and T cell exposure to SEV. Surprisingly, SEV did not alter MHC or co-stimulatory receptor expression on APCs, but caused APCs to upregulate indoleamine 2,3 deoxygenase, an enzyme known to indirectly inhibit T cells. Thus, SEV reduce the ability of APCs to activate T cells. We propose here that these immune-inhibitory properties of SEV may be intended to prevent immune responses against semen-derived antigens, but can be hi-jacked by genitally acquired viral infections to compromise adaptive cellular immunity.

## Introduction

Semen, beyond carrying spermatozoa, affects immune responses in the recipient genital tract, influencing both the establishment of pregnancy and susceptibility to infection [1–4]. Seminal

**Funding:** This work was supported by NIH grants R21 AI095023 and R01 DA040386, a Royalty Research Grant (A97523) from the University of Washington (all F.H.), an Interdisciplinary Training Grant in Cancer 2T32CA080416 (L.V.), NIH R01 AI123323 to (M.P.), T32AI007509 (V.D.), the Center for Exposures, Diseases, Genomics, and Environment (E.N.), and the Burroughs Wellcome Fund Career Award at Scientific Interfaces (E.N.).

**Competing interests:** The authors have declared that no competing interests exist.

fluid induces inflammatory cytokines and leukocyte recruitment [1, 5, 6], expansion of regulatory T cells (Treg) [7, 8] and differentiation of tolerogenic dendritic cells (DC) [8, 9]. Part of this response appears to be mediated by transforming growth factor beta (TGF-β) and prostaglandins [2, 10]. Most of these and other studies of semen have used unfractionated seminal plasma, so the contribution of specific sub-cellular components remains largely unexplored.

Semen contains one of the highest reported concentrations of extracellular vesicles of any body fluid [11–14]. The term extracellular vesicles (EV) encompasses both microvesicles, released from the plasma membrane, and exosomes, generated from endosomal multivesicular bodies. EV carry a subset of proteins, mRNAs, and non-coding regulatory RNAs including microRNAs and tRNA fragments, specifically enriched from the cell type of origin. EV in general can be taken up by specific target cells via membrane surface proteins and are recognized modulators of immune responses, capable of either stimulatory or regulatory effects [15]. Seminal EV (SEV), in particular, derived from multiple cell types in the male genital tract, have been shown to inhibit lymphoproliferative responses, phagocytosis, complement activity and natural killer cell function [12, 13, 16–19]. We have reported that SEV carry small potentially immunoregulatory RNA, which may be in part responsible for their immunosuppressive effects [11].

Given the reported immunoregulatory effects of EV, including SEV, here we investigated whether SEV impact adaptive cellular immune responses. Though there is a clear evolutionary rationale for semen to induce tolerance, maintaining resistance to sexually transmitted infections (STIs) is also essential, thus requiring a delicate balance between reactivity and tolerance [2]. We hypothesized that SEV might temporarily blunt pre-established memory immune responses, including those induced by vaccines against STIs, allowing infection to take hold and spread. Since STIs are almost always acquired in the presence of semen, and SEV are likely to penetrate the mucosa at the same locations as similarly sized viruses, this could create a particularly tough hurdle for a vaccine to overcome. We found that SEV were rapidly and efficiently taken up by peripheral and vaginal DC, but not T cells. Exposure of DCs to SEV led to marked impairment of downstream antiviral CD8+ T memory cell function. The T cell inhibitory factor indoleamine 2,3 deoxygenase (IDO), was significantly upregulated at the mRNA and protein level in SEV-exposed DCs, suggesting a mechanism by which SEV induce tolerance. These findings imply a potentially potent effect of SEV on immune function in the recipient mucosa, which could affect fertility and transmission of STIs such as human immunodeficiency, Zika, human papilloma and herpes simplex virus [1, 3, 4, 20–22].

## Materials and methods

### Human blood and semen samples

Blood samples were obtained from healthy HIV-negative men at the Vaccine Trials Unit in Seattle. Semen samples were obtained from this clinic or from the University of Washington Male Fertility Program. Written informed consent was obtained from each donor. All protocols were approved by the Institutional Review Boards of the University of Washington and the Fred Hutchinson Cancer Research Center (IR file numbers 5690 and 4323). Semen was obtained by masturbation from reproductively normal male donors 18–65 years of age. The entire ejaculates were collected in a sterile container, mixed with 3 ml of RPMI media, and kept on ice for less than 3 h prior to processing. Whole blood was collected in acid citrate dextrose tubes and processed the same day.

### Peripheral blood mononuclear cell (PBMC) purification and culturing

PBMC were isolated from whole blood by centrifugation over Lymphoprep medium according to the manufacturer's instructions (StemCell technologies) and frozen in 10% DMSO (Sigma)/

90% FBS in liquid nitrogen. For all experiments, cells were thawed and rested overnight in media before use. For DiI entry experiments and stimulations with SEB, PMA/Ionomycin, and antigenic peptides and proteins, cells were thawed and cultured in AIM-V serum free media (Thermo Fisher). For killing assays, DC were generated and experiments conducted in RPMI supplemented with 10% FBS and penicillin and streptomycin (Hyclone).

## Purification and quantification of extracellular vesicles

EV were purified from semen as previously described [11]. Briefly, following liquefaction, seminal plasma was separated from the cell fraction by centrifugation and cell debris removed by 0.45 μm and 0.22 μm syringe filtration (Millex HA). EV were purified by ultracentrifugation at 100,000 x g in a swinging bucket rotor over 20mM Tris/30% sucrose/deuterium oxide cushion (pH 7.4) for 2 h then over a 20mM Tris/25% sucrose/deuterium oxide cushion (pH 7.4) for 14 h [23]. The sucrose cushions containing EV were pooled and washed with 30 mL of Dulbecco's phosphate-buffered saline (PBS) by centrifugation at 2400 x g in an Amicon Ultracel 100 kDa cellulose centrifugal filter and concentrated to a final volume of 425 μl– 3.2 ml per donor. EV were stored at -80˚C. The SEV pool consisted of vesicles purified from 5 different semen donors.

Concentration and size distribution of EV from individual donors and from SEV pools were measured by nanoparticle tracking analysis using a Nanosight NS300 instrument (Malvern) according to the manufacturer's instructions. In brief, SEV samples were vortexed and serially diluted to a final dilution of 1:6,000–1:15,000 in filtered molecular grade $H_2O$. Blank filtered $H_2O$ was run as a negative control. Each sample analysis was conducted for 60 seconds using Nanosight automatic analysis settings. Samples were evaluated in triplicate and concentration values were averaged.

## DiI labeling and SEV entry into PBMC and vaginal LC

SEV pools were stained with DiI lipophilic tracer (ThermoFisher) at 0.5 mg/ml for 20 min at 37˚C according to the manufacturer's instructions. After staining, SEV were resuspended in 15 ml of PBS and washed and concentrated to their original volume by centrifugation at 2400 x g in an Amicon Ultracel 100 kDa cellulose centrifugal filter. The final concentration was determined by Nanoparticle Tracking Analysis (NanoSight, Malvern). Dye alone controls were generated with the same protocol using PBS in place of SEV. PBMC were incubated at 37˚C for the indicated time with $10^5$ DiI-labeled SEV per cell or the same volume of dye alone control. To test mucosal APCs, we used *ex-vivo* vaginal tissue from vaginal repair surgeries. Tissues were processed as in [24]; briefly tissue was chopped into small pieces and washed extensively to remove loose epithelial cells, then cultured without moving for 72 hours. Supernatants containing migrated cells were gently removed from the culture flasks, and cells were washed. Total cells were cultured with SEV as for PBMC, then fixed and stained for MHC class II expression and counterstained with DAPI. For annexin V inhibition of SEV binding to cellular phosphatidylserine receptors, DiI-labeled SEV were incubated with 15 μg/ml of purified annexin V (BD Biosciences) in 1x annexin binding buffer (0.01 M Hepes pH 7.4; 0.14 M NaCl; 2.5 mM $CaCl_2$) at room temperature for 20 minutes before adding SEV to cells. For cytochalasin D inhibition of phagocytosis, cells were pre-treated with 20 μM cytochalasin D (Sigma) for 30 minutes at 37˚C, then washed in media and incubated with DiI-labeled SEV. Samples were moved to 4˚C at each timepoint, until the end of the experiment when all samples were stained for flow cytometry analysis. Cells were stained with LIVE/DEAD Fixable Aqua Dead Cell Stain Kit (ThermoFisher) according to the manufacturer's instructions,

**Table 1. Antibodies for PBMC staining.**

| Marker | Color | Company | Resource Identification Portal # (RRID) |
|--------|-------|---------|------------------------------------------|
| HLA-DQ | FITC | BD Biosciences | AB_400304 |
| CD14 | PE-Cy7 | BD Biosciences | AB_396848 |
| CD8 | BUV395 | BD Biosciences | AB_2722501 |
| CD11c | APC | BD Biosciences | AB_398680 |
| CD19 | APC/A750 | Beckman Coulter | AB_2728101 |
| CD3 | ECD | Beckman Coulter | AB_130860 |
| CD4 | BV785 | BioLegend | AB_2561365 |

followed by surface staining, using the antibodies listed in Table 1. Samples were acquired on a BD LSRII (BD Biosciences) and analyzed using FlowJo software.

## PBMC stimulation

PBMC were thawed and rested overnight, then washed and suspended at $2.5 \times 10^6$ cells/ml—$5 \times 10^6$ cells per ml. Experiments were done with 200 μl of cells per well in 96 well U bottom plates. SEV at $10^5$ per cell were added to the cells at the same time as stimulations. The following reagent was obtained through the AIDS Reagent Program, Division of AIDS, NIAID, NIH: HCMV pp65 Peptide Pool (Catalog # 11549). This is a pool of cytomegalovirus (CMV) 15-mer peptides overlapping by 11 amino acids spanning the entire pp65 protein. 43 Epstein-Barr Virus (EBV) peptides developed to stimulate both CD4+ and CD8+ T cells were purchased from Miltenyi Biotech (EBV PepTivator consensus peptides). Peptides were used at 1 μg/ml per peptide. CMV-infected cell lysate and EBV-infected cell lysate (EastCoast Bio) were used at 10 μg/ml. Staphylococcal enterotoxin B (SEB; Sigma) was used at 1 μg/ml, Phorbol 12-myristate 13-acetate (PMA; Sigma) at 50 ng/ml and Ionomycin (Sigma) at 1 μg/ml. Peptide diluent (1% DMSO) was added to negative control wells. All stimulations included Brefeldin A (10 μg/ml, Sigma) and the co-stimulatory antibodies CD28 and CD49d (each at 1 μg/ml; BD Biosciences); experiments assessing CD107a expression also included 0.133 μl/well of Golgistop (BD Biosciences). For negative control wells all reagents were present except viral proteins or peptides, and equivalent concentrations of DMSO antigen diluent were included. Stimulations were incubated at 37˚C for 6 h. T cell responses were measured using a protocol developed by the HIV Vaccine Trials Network [25, 26]. Briefly, cells were stained with LIVE/DEAD Fixable Aqua Dead Cell Stain Kit and then fixed, permeabilized, and stained with the reagents in Table 2. For experiments assessing CD107a surface expression, cells were incubated overnight at 4˚C with anti-CD107a antibody before live/dead cell staining. Samples were acquired on a BD LSRII and analyzed using FlowJo software. All assays were done in duplicate and responses were averaged in each experiment. Each individual's blood specimen was analyzed in at least

**Table 2. Antibodies for intracellular cytokine staining.**

| Marker | Color | Company | Resource Identification Portal # (RRID) |
|--------|-------|---------|------------------------------------------|
| CD4 | PE-Cy5 | BD Biosciences | AB_395753 |
| CD3 | ECD | Beckman Coulter | AB_130860 |
| CD8 | APC-Cy7 | BD Biosciences | AB_396892 |
| IL-2 | PE | BD Biosciences | AB_397231 |
| TNF-α | Alx-700 | BD Biosciences | AB_396978 |
| IFN-γ | v450 | BD Biosciences | AB_1645594 |
| CD107a | PE-Cy7 | BD Biosciences | AB_10644018 |

**Table 3. Antibodies for DC phenotyping.**

| Marker | Color | Company | Resource Identification Portal # (RRID) |
|--------|-------|---------|------------------------------------------|
| CD40 | PE | Beckman Coulter | AB_131623 |
| CD80 | PE-Cy5 | BD Biosciences | AB_397239 |
| HLA-DQ | FITC | BD Biosciences | AB_400304 |
| CD86 | BV421 | BD Biosciences | AB_11153866 |
| CD83 | BUV737 | BD Biosciences | AB_2738809 |
| CD14 | PE-Cy7 | BD Biosciences | AB_396848 |
| IDO | PerCp-eFluor | eBioscience | AB_2573887 |

two separate experiments. The fraction of CD4+ or CD8+ T cells producing any combination of IFN-γ, TNF-α, or IL-2 were calculated using Boolean gates. Positive responses were defined as two-fold above averaged background derived from negative control wells. Raw data including production of each individual cytokine is in S1 File.

## Generation of monocyte-derived dendritic cells (DC)

DC were generated from monocytes using a standard protocol [27]. Briefly, monocytes were isolated by plastic adherence and cultured for 5 days in the presence of 800 IU/ml GM-CSF and 250 IU/ml IL-4 (Peprotech). On day 5, non-adherent DC were harvested, loaded with peptide antigens and SEV, and matured overnight. CMV peptides were used at 1 μg/ml, SEV were added at $10^5$ per cell, and the maturation stimulus was monocyte-conditioned media mimic (MCM mimic), consisting of a final concentration of 50 ng/ml TNF-α, 50 ng/ml IL-1β, 1.5 μg/ml IL-6 and 10 μg/ml prostaglandin E2 (all Peprotech). DC phenotype for some experiments was confirmed by assessing cells using flow cytometry (high HLA-DQ, low or absent CD14, moderate CD80). Matured, antigen-loaded DC were washed before use in experiments.

## Phenotyping and qPCR analysis of SEV-treated DC

Expression of co-stimulatory markers on DC was analyzed by flow cytometry. DC were treated with SEV or mock treated overnight, washed, stained with LIVE/DEAD Fixable Yellow Dead Cell Stain Kit (ThermoFisher), stained with the primary antibodies listed in Table 3, fixed with 4% PFA and analyzed by flow cytometry. For expression of indoleamine 2,3 deoxygenase (IDO), following viability staining, cells were stained with anti-HLA-DQ, then washed and fixed and permeabilized using Fixation/Permeabilization Solution Kit (BD Biosciences) according to the manufacturer's instructions. Permeabilized cells were stained with anti-IDO antibody for 30 minutes, washed, and fixed in 1% PFA prior to acquisition on a BDLSRII and analysis using FlowJo 10.

For measurement of IDO, IL-10, TGF-β, TNF-α, and cyclophilin A (PPIA) by quantitative PCR, a minimum of 60,000 DC were SEV or mock treated overnight, then washed, and RNA was isolated using the RNeasy Plus Mini kit (Qiagen) according to the manufacturer's instructions. Random-hexamer primed cDNA was generated from 200 ng input RNA using the High-Capacity cDNA Reverse Transcription Kit (Ambion) according to the manufacturer's instructions. cDNA was diluted 1:10 with water and 5 μL was used in 15 μL final quantitative PCR reaction using PrimeTime Gene Expression Master Mix (Integrated DNA Technologies) and primer probe sets specific for each gene (Table 4). Thermocycling was done on a Quant-Studio 5 (Applied Biosystems). Fold regulation of each gene was assessed compared to the reference PPIA gene and mock treated control DC cultures using the delta-delta Ct method [28].

**Table 4.  Primer/Probe Sequences and Sources for qPCR. IDT is integrated DNA technologies.**

| Assay | Company | Forward primer | Reverse primer | Probe |
|---|---|---|---|---|
| PPIA | IDT Hs. PT.58v.38887593.g | CAAGACTGAGATGCACAAGTG | GTGGCGGATTTGATCATTTGG | /5TET/AATTCACGC/ZEN/AGAAGGAACCAGACAGT/ 3IABkFQ |
| IDO | IDT Hs.PT.58.924731 | ACGTCCATGTTCTCATAAGTCAG | CCTTACTGCCAACTCTCCAA | /56FAM/CCAGTTTGC/ZEN/CAAGACACAGTCTGC/ 3IABkFQ/ |
| TNFα | ThermoFisher Hs00174128_m1 | | | |
| IL-10 | IDT Hs.PT.58.2807216 | TCACTCATGGCTTTGTAGATGC | GCGCTGTCATCGATTTCTTC | /56FAM/AGGCATTCT/ZEN/TCACCTGCTCCACG/ 3IABkFQ |
| TGF-β | IDT HS.PT.58.39813975 | GTTCAGGTACCGCTTCTCG | CCGACTACTACGCCAAGGA | /56-FAM/ACCCGCGTG/ZEN/CTAATGGTGGAA/ 3IABkFQ/ |

## Stimulating DC and T cells separately

T cells were negatively isolated from PBMC using the Human Pan T Cell Isolation Kit (Miltenyi Biotech) according to the manufacturer's instructions. DC were stained with CFSE before antigen loading overnight to identify and exclude during flow cytometry analysis any T cells surviving in DC cultures. Isolated T cells were exposed to $10^5$ SEV per cell overnight before washing and mixing with matured, antigen-loaded, CFSE-stained DC from the same individual at a 30:1 ratio of T cells to DC. Co-cultures proceeded for 6 h before analyzing T cells for production of IFN-γ, TNF-α or IL-2 as described for PBMC stimulations. For experiments where only fractions of DC were exposed to SEV, all cells were loaded with CMV peptide antigens and matured, and then cultures were split and only 50% or 20% of the DC were exposed to SEV at $10^5$ per cell. SEV-exposed and -unexposed DC were washed and then combined before mixing with responder T cells.

## Degranulation and killing assay

Optimal detection of cell surface CD107a requires that the anti-CD107a antibody be present during T cell stimulation [29]. However, because EV in general also express CD107a, the presence of SEV decreased the amount of CD107a detectable on the cell surface. Consequently, for these experiments, a stimulus that induced strong CD107a expression (SEB) was used, and the anti-CD107a antibody was not added until after the cells had been stimulated for 6 h and the SEV washed off. Cells were incubated overnight at 4˚ with anti-CD107a PE-Cy7 (BD Biosciences) before staining as described above.

To generate antigen-specific cytotoxic CD8+ T cells, CMV peptide-loaded and matured DC were mixed with pan T cells isolated from the same volunteer at a 20:1 ratio of T cells to DC and expanded for 6 days. Negative control CD8+ T cells were generated by culturing with DC that had not been exposed to antigen. During this time, a second set of monocyte-derived DC were cultured for 5 days as above and split upon harvesting. One half of DC were stained with 5 μM CFSE for 8 minutes at 37˚, and the second half stained with CellTrace Violet (CTV) according to the manufacturer's instructions (both ThermoFisher). CFSE-stained DC were loaded with irrelevant HIV-1 Gag peptides (obtained through the AIDS Reagent Program, Division of AIDS, NIAID, NIH: HIV-1 Consensus A Gag Peptide Pool) to serve as targets for non-specific killing, and CTV-stained DC were loaded with CMV peptides to serve as targets for specific killing (both at 1 μg/ml). Both sets were matured overnight with MCM mimic as described above. In conditions where targets were treated with SEV, $10^5$ SEV per DC were added to both sets of DC at the same time as peptides and MCM mimic. The next day CD8+ T cells were isolated from expansion cultures using the Human CD8+ T Cell Isolation Kit

(Miltenyi Biotech) according to the manufacturer's instructions. DC were washed, counted and 25,000 of each CTV- and CFSE-stained (50,000 total DC) were mixed with CD8+ T cells at a 10:1, 5:1, or 2.5:1 ratio of CD8+ T cell to DC. In some conditions, SEV at $10^5$ per total cell were added at the time of mixing CD8+ T cells and DC. Killing proceeded for 6 h before stopping the experiment, staining with LIVE/DEAD Fixable Near-IR Dead Cell Stain Kit (Thermo-Fisher) and fixing in 1% PFA. Samples were acquired on a BD LSRII and analyzed using FlowJo software. The ratio of CTV-containing specific target DC to CFSE-containing non-specific target DC in the live cell-gated population was calculated. Specific killing was calculated as 1 –[(ratio of CTV:CFSE DC for experimental condition) / (ratio of CTV:CFSE DC for controls with no CD8+ T cells)] x 100.

## Statistics

All statistics were calculated using Prism (version 6.05, GraphPad). Cytokine production in SEV-treated or -untreated stimulated samples were averaged separately for each blood specimen for CD4+ and CD8+ T cells across experiments and compared by Wilcoxon matched-pairs signed rank test. Percent reductions in SEV treated samples were compared across treatments using unpaired two-sided t-tests, or, when there were 3 conditions, one-way ANOVA with repeated measures.

## Results

### Antigen-presenting cells but not T cells bind and take up SEV by a phosphatidylserine-dependent mechanism

SEV were purified and quantified by nanoparticle tracking analysis as previously reported [11]. SEV from 5 different donors were combined into pools and labeled with the lipophilic dye DiI. To determine which types of leukocytes bind to or internalize SEV, PBMC were incubated with labeled SEV ($10^5$ vesicles per cell) for up to 10 h. This dose of SEV was chosen based on our estimates of the number of cells in the surface area of an adult human vagina, taking into consideration published measurements of the size of the vaginal cavity (average 110 $cm^2$) [30, 31], the size of vaginal epithelial cells (64 μm diameter) [32], and the very high vesicle concentration in semen (average 1.98 x $10^{13}$ per ejaculate). We calculate that cells in the human vagina could be exposed to 5.8 x $10^6$ SEV each, but with considerable variation due to an uneven distribution of semen and due to microabrasions where semen can more easily penetrate the mucosa. Notably, the similar size distribution of exosomes and many viruses (Fig 1A) indicates that their mucosal penetration routes are likely to overlap. Therefore, $10^5$ SEV per cell is a physiologically reasonable dose and is used in all subsequent experiments unless otherwise noted.

Flow cytometric analysis showed that most human monocytes and dendritic cells (DC) bound or internalized SEV, and did so in a time-dependent manner, while only a small fraction (<10%) of T or B cells bound to or took up SEV (Fig 1B & 1C). Blood-derived DC, and vaginal Langerhans cells (LC) and T cells, were also incubated with labeled SEV and assessed by confocal microscopy. While blood DC and vaginal LC showed punctate DiI staining, indicating vesicle binding or internalization, vaginal T cells never became positive for DiI (Fig 1D).

EV contain exposed phosphatidylserine (PS), [33–36] and APCs express numerous PS receptors [37–39]. To determine whether PS plays a role in the uptake of SEV by APCs, PS on SEV was blocked with annexin V, which binds to phosphatidylserine, before incubation with PBMC. Pre-treatment of SEV with annexin V substantially reduced binding and uptake (Fig 1D). This indicates that molecules on the surface of SEV are important for recognition and

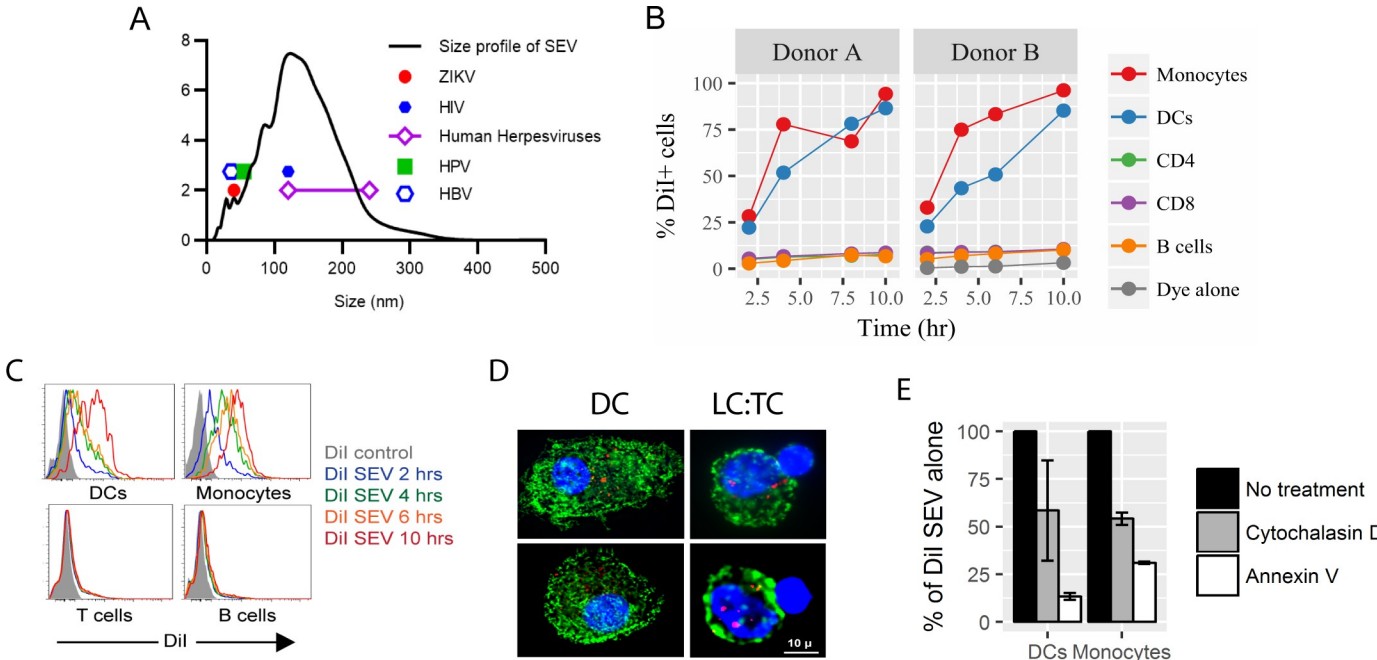

**Fig 1. Size profile of SEV and binding of SEV to leukocytes.** (A) Pooled preparations of SEV were analyzed using nanoparticle tracking analysis. A representative size profile is presented along with the published sizes of common sexually transmitted viruses to demonstrate that SEV and viral STIs substantially overlap in size. Sexually transmitted human herpesviruses include EBV, CMV, HSV-1, HSV-2, and KSHV. (B) Fluorescently labeled SEV were incubated with PBMCs at $10^5$ per cell. At 2, 4, 6 or 8, and 10 hours post addition samples were moved to 4° to arrest SEV binding and entry. The next day, samples were stained for phenotypic markers and analyzed by flow cytometry. Data from two different PBMC donors are shown as the percent of cell type positive for DiI. (C) Histograms of DiI staining on PBMCs gated on the indicated populations. (D) Confocal microscopy of blood dendritic cells (DC) or Langerhans cells (LC) from vaginal mucosal tissue exposed to DiI-labeled SEV for 4 hrs. DC or LC were stained for expression of MHC class II (HLA-DQ) and are shown in green, DiI SEV in red, and cell nuclei in blue (DAPI). Nuclei of unstained cells conjugated to LCs are T cells. (E) Annexin and cytochalasin D inhibit SEV binding and entry. Annexin-pretreated SEV (10 μg/mL) were incubated with PBMCs; untreated DiI-labeled SEV were added to cytochalasin D-pretreated (20 μM) or untreated PBMCs as above. Data are presented as percent of cells positive for DiI-SEV relative to the condition without annexin or cytochalasin D treatment, averaged across two independent PBMC donors.

uptake by APCs, either via exposed PS binding to PS receptors or other molecules that are sterically blocked by annexin V. Additionally, treatment of PBMC with cytochalasin D, an inhibitor of phagocytosis, blocked around 50% of SEV detection in blood APCs (Fig 1D). Because cytochalasin D should block only internalization of SEV, not binding to cells, this implies that a large fraction of SEV were actively internalized by these cells.

## SEV impair antigen-specific T-cell responses

The reported immunosuppressive effect of unfractionated seminal plasma [40], viewed in conjunction with our published data demonstrating potentially immunoregulatory RNA molecules in SEV [11] and the likelihood that SEV could penetrate the vaginal mucosa and bind to APCs, led us to ask whether SEV impair adaptive immunity. Specifically, we tested T cell recall responses to antigenic cytomegalovirus (CMV) or Epstein-Barr virus (EBV) peptides within whole PBMC cultures containing APCs as well. We also tested T cell responses to whole protein antigens derived from lysates of CMV- or EBV-infected cells to assess the significance of protein processing by APCs. PBMC from normal healthy volunteers were stimulated for 6 h with the peptides or protein lysates. Memory immune responses were assessed by intracellular cytokine staining for production of IFN-γ, IL-2 and/or TNF-α by CD4+ and CD8+ T cells. Results for individual cytokines are included as S1 File. The fraction of cells responding varied considerably for each antigen and most volunteers did not have both CD4+ and CD8+ T cell

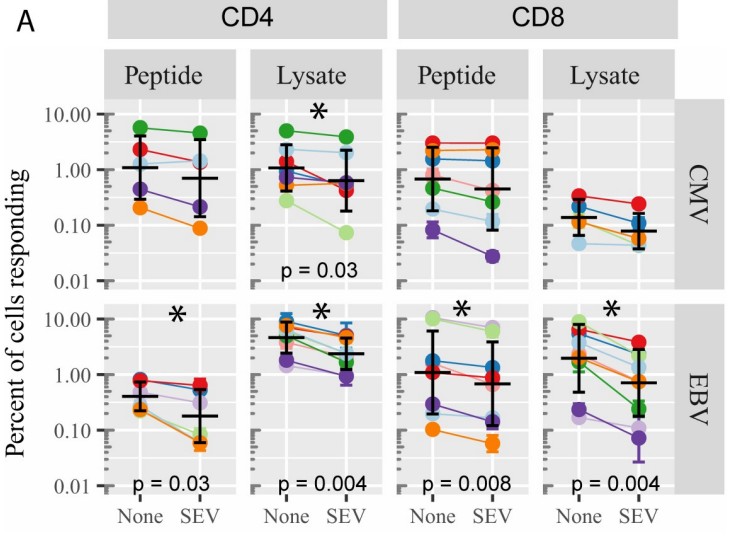

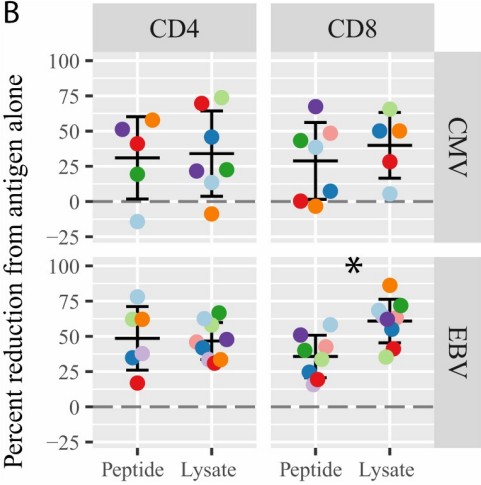

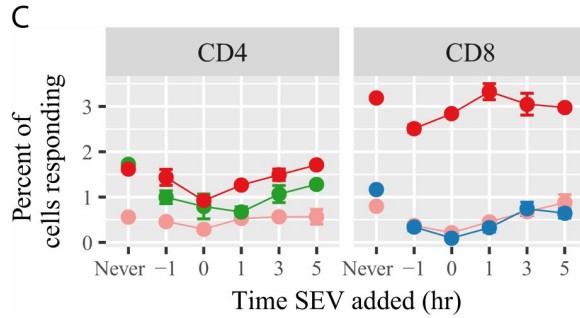

**Fig 2. SEV inhibit EBV and CMV-specific memory immune responses.** (A) PBMC from 10 individuals were exposed to CMV peptides, CMV lysate, EBV peptides or EBV lysate and $10^5$ SEV per cell (or left unexposed). Production of cytokines was assessed by intracellular cytokine staining for IFNγ, IL-2 and TNFα, and the sum of the percent of cells responding with any cytokine is reported, separately for CD4+ and CD8+ T cells. Each color indicates a different blood donor. Each donor was tested in 2–3 independent experiments and responses are averaged. Significance by Wilcoxon matched-pairs signed rank test. (B) For each donor, the average percent reduction in the fraction of cytokine producing cells from antigen alone-exposed cells is plotted, separately for CD4+ and CD8+ T cells. Colors indicate different donors, as in (A). Significance by unpaired t test. (C) PBMC were exposed to CMV peptides at t = 0 and $10^5$ SEV per cell were added at the indicated times. "Never" indicates no addition of SEV. Data were analyzed for cytokine production as in (A).

responses to all four antigens, thus not all donors appear in each panel. Analyzing all detectable T cell responses, SEV significantly impaired cytokine production in response to CMV lysate, and EBV peptide and lysate, in CD4+ T cells, and to EBV peptide and lysate in CD8+ T cells (Fig 2A). In CD4+ T cells, responses were impaired to nearly the same extent for peptide and lysate (mean reduction 31.0 ±13.1% SEM for CMV peptide and 34.0 ±11.5% SEM for CMV lysate; 48.6 ±9.2% SEM for EBV peptide and 46.7 ±4.4% SEM for EBV lysate). In CD8+ T cells, responses were more impaired in response to lysate than peptide (mean reduction 28.8 ±10.3% SEM for CMV peptide compared to 39.87 ±10.4% SEM for CMV lysate; 35.6±5.3% SEM for EBV peptide compared to 60.8±5.2% SEM for EBV lysate, p = 0.004) (Fig 2B). SEV-mediated reductions in cytokine production depended on the time SEV were added to the cells. Adding SEV to the PBMC cultures at the same time (time zero) or within 1 h before or after adding CMV antigen resulted in the greatest reduction of cytokine production by T cells. However, in some cases, the presence of SEV for 3 h or even just 1 h at the end of the 6 h assay still impaired cytokine production (Fig 2C). These results demonstrate that SEV markedly

impair CD4+ and CD8+ T cell responses to recall antigens. Notably, CD8+ but not CD4+ T cell responses are more strongly compromised when more extensive antigen processing by APCs is required.

## SEV-mediated immunosuppression occurs upstream of protein kinase C activation

The selective uptake of SEV by APCs (Fig 1), as well as greater impairment of T cell activation when more extensive antigen processing is required (Fig 2), lead us to investigate what T cell activation pathways were affected by the presence of SEV. We compared the effect of SEV on T cell activation following stimulation with the superantigen staphylococcus enterotoxin B (SEB), which activates T cells by cross-linking MHC on APCs and the T cell receptor [41], with stimulation by PMA/ionomycin, which directly activates protein kinase C in T cells, negating the need for T cell receptor stimulation [42]. SEV significantly inhibited T cell responses to both SEB and PMA/ionomycin, but the magnitude of reduction was much larger for SEB (32.5% ±4.2 SEM for CD4 and 37.4% ± 5.0 SEM for CD8) than for PMA/ionomycin (16.2% ±4.4 SEM for CD4 and 16.2% ±4.5 SEM for CD8) (Fig 3A and 3B, individual cytokine results in S2 F). This indicates that SEV exert their most potent immunosuppression when APCs and T cells interact, but that T cell activation in generally is at least partially impacted by the presence of SEV in mixed PBMC cultures.

## Exposing only APCs to SEV recapitulates impaired CD8+ T cell responses

The finding that SEV act, at least in part, through APCs rather than directly on T cells led us to ask whether exposing only APCs, rather than mixed PBMCs, to SEV would recapitulate the impaired T cell responses we observed. Autologous monocyte-derived DC and T cells were cultured separately and exposed to CMV peptide with or without SEV overnight. Cells were then washed to remove free SEV and mixed 30 T cells to 1 DC for 6 h before assessing T cells for cytokine production. For CD4+ T cells, exposing separate fractions of cells to SEV never caused the same level of inhibition of cytokine production as seen in mixed PBMC cultures (Fig 3C). In contrast, CD8+ T cell cytokine production was impaired similarly when only the DC were exposed to SEV compared to exposing mixed PBMC cultures (Fig 3C). Treating only the CD8+ T cells with SEV had little to no inhibitory effect (Fig 3C). Thus, SEV impair the activation of CD8+ T cells chiefly at the level of APCs, while inhibition of CD4+ T cell activation is more dependent on SEV presence during the APC-CD4+ T cell contact phase.

*In vivo*, due to the uneven distribution of semen after ejaculation, it is most likely that a moderate number of APCs would be exposed to a very high dose of SEV, while many APCs would remain unexposed. Therefore, we did experiments exposing 100%, 50% or 20% of CMV-loaded DC to SEV (n = 3 individuals). Exposing 100% of DC to SEV led to robust inhibition of CD8+ T cell activation, but exposing only 50% or 20% of DC to SEV still had a strong immunosuppressive effect (Fig 3D). This suggests that not all APCs in the mucosa must be exposed to SEV to cause meaningful immunosuppression.

## Exposure of APCs to SEV also impairs the subsequent degranulation and killing capacity of memory CD8+ T cells

Up to this point, we had defined T cell responses by antigen-specific cytokine production and were now interested to know whether SEV-mediated inhibition extends to suppressing cytotoxicity, a prime effector mechanism of CD8+ T cells. First, we assayed CD8+ T cell degranulation by measuring the appearance of CD107a (LAMP-1), a marker for degranulation, on the

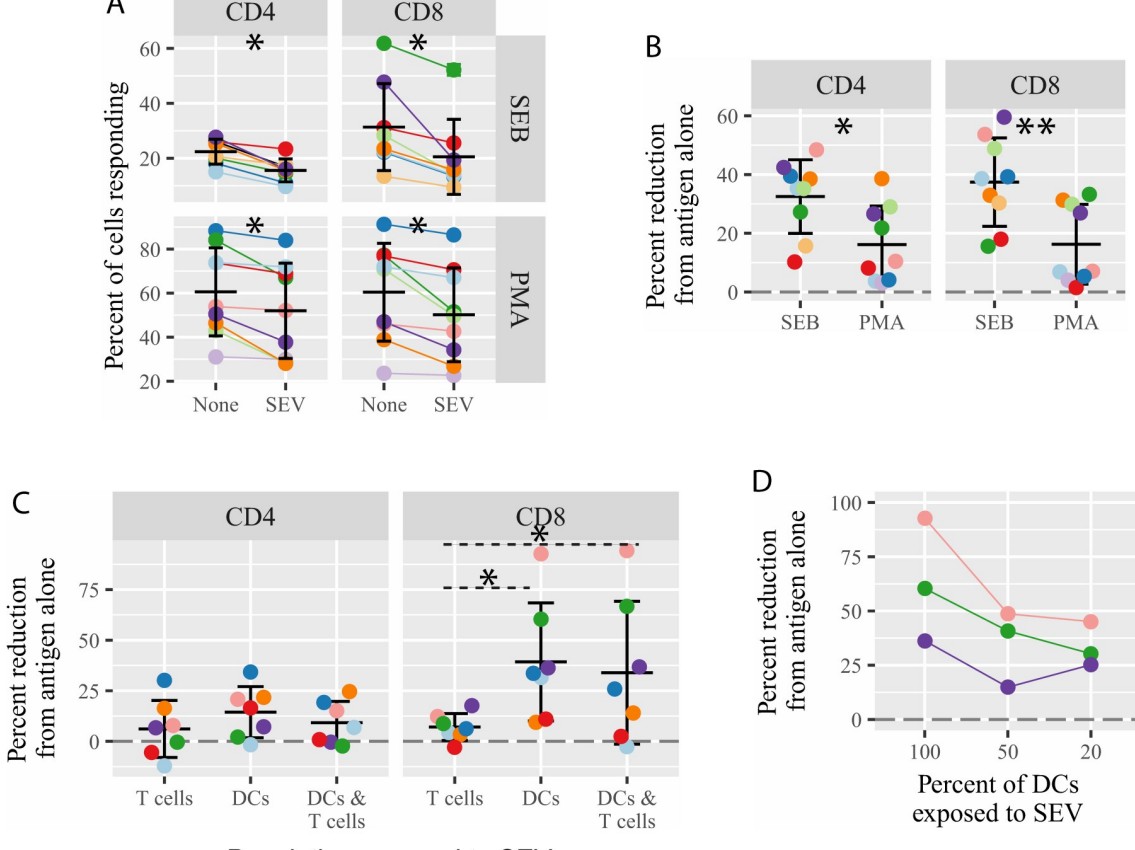

**Fig 3. SEV inhibit T cell cytokine production upstream of protein kinase C activation.** (A) PBMC were stimulated with staphylococcal enterotoxin B (SEB) at 1 μg/mL or phorbol myristate acetate (PMA) at 50 ng/ml plus ionomycin at 1 μg/mL for 6 hrs, in the presence or absence of SEV. Production of cytokines was assessed by intracellular cytokine staining for IFNγ, IL-2 and TNFα, and the sum of the percent of cells responding with any cytokine is reported. Each color indicates a different blood donor. The reduction in the fraction of cytokine-producing cells in SEV-exposed cells is significant by Wilcoxon matched-pairs signed rank test (CD4+ SEB p = 0.0078; CD4+ PMA p = 0.0039; CD8+ SEB p = 0.0078; CD8+ PMA p = 0.0039). (B) The percent reduction in SEV-exposed cells from stimulated cells alone is plotted. Differences in the percent reduction for SEB compared to PMA/ionomycin are significant by unpaired t test (CD4+ p = 0.02; CD8+ p = 0.006). (C) DC differentiated from blood precursors were exposed to CMV peptides and a maturation stimulus. SEV were added to DC or autologous purified T cells or both at the time of antigen loading and incubated for 20 hrs. After washing, T cells were mixed with antigen-loaded DC at a ratio of 1 DC to 30 T cells. Percent reduction in the sum of cytokine-producing cells compared to CMV-loaded DC mixed with SEV-untreated T cells is plotted. Significance by one-way ANOVA with repeated measures (p = 0.0003). (D) DC were exposed to CMV peptides and 50% or 20% of the DC were simultaneously treated with $10^5$ SEV/cell. DC were washed, SEV-treated and -untreated DC were combined, and the DC mixtures were added to autologous T cells at a DC to T cell ratio of 1:30. Production of cytokines was assessed as in (C) and percent reduction from antigen-loaded, SEV-unexposed DC + T cells is plotted.

cell surface after PBMC stimulation with SEB in the presence or absence of SEV [29, 43]. SEV significantly impaired degranulation, reducing the fraction of CD8+ T cells expressing surface CD107a by 33.7% (Fig 4A), similar to the reduction seen for cytokine production by SEB-stimulated CD8+ T cells (35.4%, Fig 4B). In some volunteers, SEB also stimulated CD107a expression on a small fraction of CD4+ T cells (Fig 4A, note different scale). Again, the presence of SEV impaired CD107a expression on these CD4+ T cells to a similar extent (30.49%) as the level of suppression seen for cytokine production (39.77%) (Fig 4B). The presence of SEV during T cell activation therefore inhibits more than one type of T cell effector mechanism.

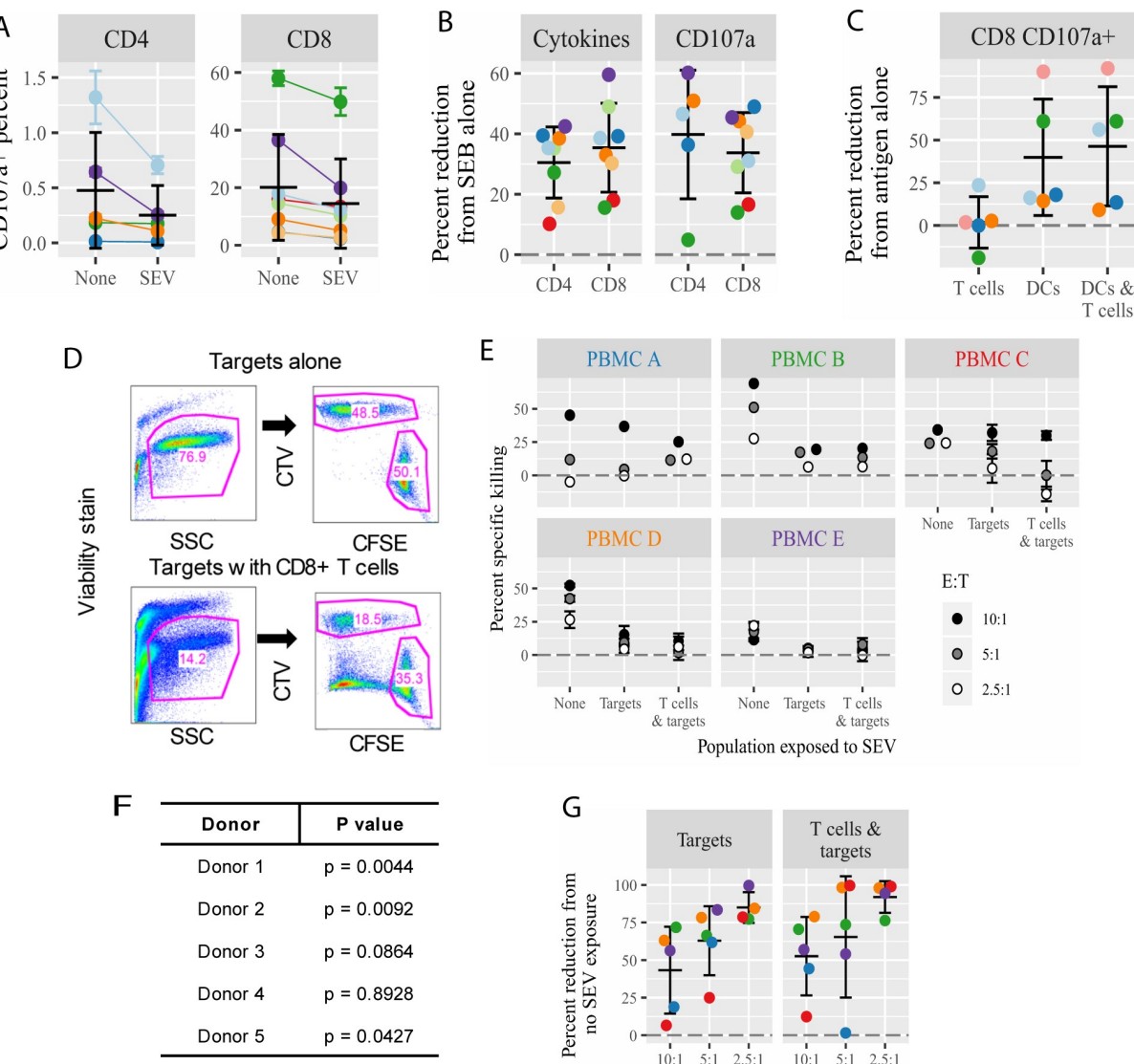

**Fig 4. CD8+ T cells activated by SEV-exposed APCs are impaired in degranulation and killing capacity.** (A) PBMC were exposed to SEB and SEV (or left unexposed) for 6 hrs and washed to remove free SEV. Degranulation of CD8+ T cells was assessed by surface staining of cells with anti-CD107a antibody and flow cytometry (gated on CD3+ T cells); percent of CD4+ and CD8+ T cells positive for CD107a is shown. Significance by Wilcoxon matched-pairs signed rank test (p = 0.008). (B) Comparison of percent reduction in the production of cytokines compared to degranulation in SEV-exposed T cells (relative to SEV-unexposed cells seeing only SEB). (C) CMV-loaded DC, T cells, or both, were separately exposed to SEV for 20 hrs before co-culturing the cells for 6 hrs, as in Fig 5C. Cell surface expression of CD107a was analyzed as in (A) and the percent reduction in CD107a+ CD8+ T cells from the no SEV condition is shown. (D) Gating scheme for killing assay. CMV-specific T cells were expanded for 6 days on CMV-peptide-loaded or unloaded DC. CD8+ T cells were then isolated by negative selection and mixed at 10:1, 5:1, and 2.5:1 ratios with autologous target DC. Target DC consisted of an equal 1:1 mix of CellTrace Violet (CTV)-stained DC loaded with CMV peptides and carboxyfluorescein succinimidyl ester (CFSE)-stained DC loaded with irrelevant HIV peptides. Cells were gated on live high side scatter target cells. Specific killing was calculated as (1 −(CTV/CFSE live cells for experimental condition / CTV/CFSE live cells for target alone control)) x 100 (E) Percent specific killing for each PBMC donor tested, calculated as in D. Non-specific CD8+ T cells were cultured with DC not exposed to any antigen but otherwise generated in the same way as CMV-specific CD8+ T cells. For "Targets", SEV at $10^5$ per cell were added to labeled DC targets at the time of antigen-loading and DC targets were washed before mixing with CMV-specific CD8+ T cells. For "T cells & targets", additional SEV were added at the time of mixing CMV-specific CD8+ T cells with labeled DC targets. E:T signifies the ratio between CD8+ T cell effector cells and DC target cells. (F) P values from comparison between SEV-treated and -untreated conditions in the killing assay in panel E, by one-way ANOVA with repeated measures. (G) The percent reduction by SEV in specific killing is plotted relative to CMV-specific CD8+ T effector cells and DC target cells not exposed to SEV. There was no specific killing at the 2.5 ratio for donor A, hence no percent reduction is plotted.

We also assessed CD8+ T cell degranulation in experiments where CMV peptide-loaded DC and T cells were separately exposed to SEV prior to washing and mixing. Very few volunteers had detectable CD107a on CD4+ T cells in response to CMV-loaded DC, so they were not analyzed in this experiment. For CD8+ T cells, exposing only DC, or DC and T cells, but not T cells alone, to SEV resulted in a reduction of degranulating CD8+ T cells (Fig 4C). This result is analogous to the cytokine results in Fig 2D and provides further evidence that SEV impair APCs directly, resulting in downstream attenuated CD8+ T cell activation and function.

In addition to degranulation markers, we measured actual antigen-specific killing by memory CD8+ T cells. T cells were cultured for 6 days with CMV peptide-loaded or -unloaded DC. CD8+ T cells were isolated from DC-T cell co-cultures by negative selection and mixed at various ratios with target DC loaded with either CMV peptides or irrelevant HIV peptides. Lysis of target DC was detected and quantified as specified in Fig 4 legend and in the Methods. Effector CD8+ T cells expanded against CMV-loaded DC robustly killed CMV-loaded target DC in a ratio dependent manner (Fig 4E). When target DC were treated with SEV during antigen loading and washed before incubation with the CMV-specific effector CD8+ T cells, the targets' killing was significantly decreased in 3 of 5 volunteers tested (Fig 4E and 4F). Adding SEV during the killing phase as well as to target DC during antigen loading further decreased the targets' killing in only one volunteer (volunteer C) (Fig 4E). Comparing the percent reduction in killing between untreated and SEV-treated targets revealed that killing was most highly reduced at lower, i.e., more physiological, ratios of CD8+ T cells to targets (Fig 4G). Taken together, these results demonstrate that exposure of APCs to SEV inhibits pathways necessary for recognition by and activation of CD8+ T cells, during both the stimulation and effector phases of adaptive T cell immunity.

## Variability of SEV-mediated T cell inhibition between individuals

Our SEV inhibition assays showed large variability between the individuals tested. Having performed several different functional assays in each SEV recipient gave us the opportunity to assess whether some people are intrinsically more susceptible to SEV inhibition than others. Indeed, some individuals showed consistently strong impairment of their T cell responses by SEV no matter what functional assay was used (e.g., individuals H and E in Fig 5A). In contrast, one person was comparatively refractory to SEV inhibition across all assays tested (individual C in Fig 5A). Overall, the difference in impairment of CD8+ T cell responses mediated by SEV between individuals was significant by one-way ANOVA (Fig 5A) and the susceptibilities to decreased cytokine production and reduced CD107a expression were correlated (Pearson r = 0.724) (Fig 5B). These results indicate that at least some individuals possess high intrinsic susceptibility to SEV-mediated inhibition of their T cell responses, whereas others may be mostly refractory.

## SEV-exposed DC do not alter expression of classical co-stimulatory molecules but upregulate the immunosuppressive enzyme indoleamine 2,3-dioxygenase

Given the evidence that SEV-exposed APCs were deficient in activating CD8+ T cells, we tested whether the expression of MHC or classical co-stimulatory molecules on APCs was impaired by SEV. We assessed expression of HLA-DQ (MHC class II), HLA-ABC (MHC class I) and the co-stimulatory molecules CD40, CD80, CD83, and CD86 in SEV-exposed monocyte-derived DC. While, as expected, addition of a maturation cocktail to immature DC

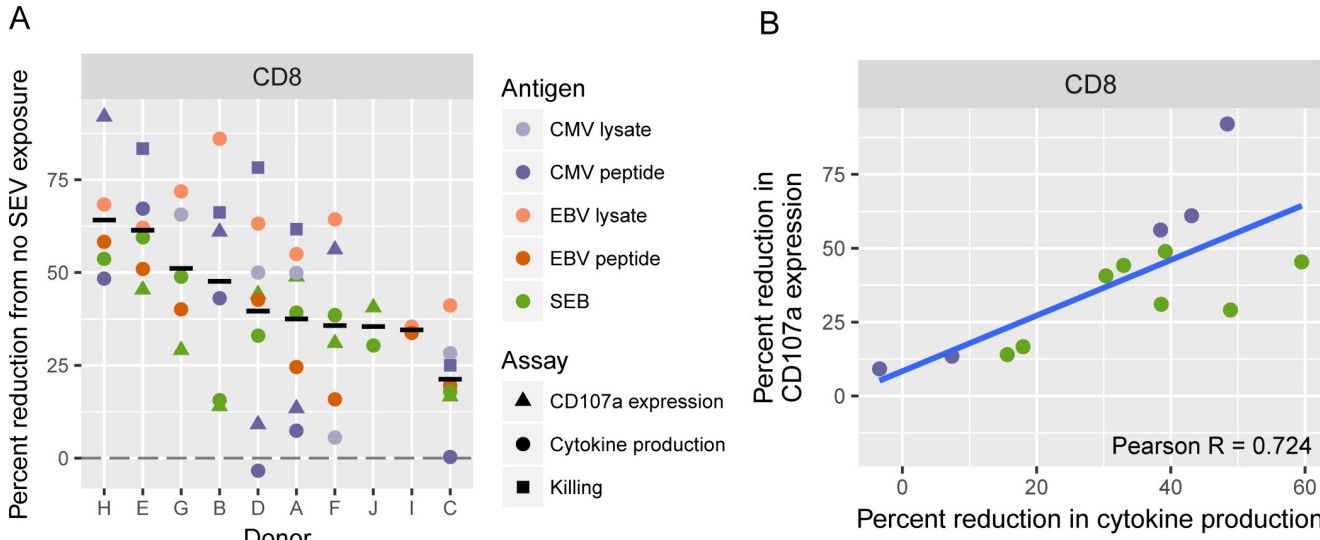

**Fig 5. PBMC donors vary in susceptibility to SEV-mediated impairment of CD8+ T cell responses.** (A) The percent reduction from control (no SEV) is plotted by PBMC donor for all tested assays. Black horizontal lines indicate mean percent reduction. Differences between donors is significant by one-way ANOVA (p = 0.024). (B) Correlation between reduced cytokine production and reduced degranulation as measured by CD107a expression on SEV-exposed CD8+ T cells. Colors as in (A). Gray areas indicate the 95% confidence interval for the correlation.

resulted in the upregulation of all markers, addition of SEV to immature DC or during maturation of DC did not alter the expression of any marker tested (Fig 6A).

Besides directly activating T cells via MHC/peptide and co-stimulatory receptor engagement, DC possess T cell immunomodulatory properties, for instance by altering expression of the enzyme indoleamine 2,3-dioxygenase (IDO) or the immunosuppressive cytokines TGF-β and IL-10. IDO catalyzes degradation of the essential amino acid tryoptophan and its overexpression leads to a tryptophan-starved microenvironment that suppresses T cell effector functions [44]. IL-10 and TGF-β can be produced by tolerogenic APCs and act in a autocrine manner to induce IDO expression or to directly suppress T cell function [45–48]. We therefore tested IDO, IL-10 and TGF-β expression in SEV-exposed DC at the level of transcriptional control (RNA expression). We did not observe any upregulation of IL-10 or TGF-β, though transcripts were present (Ct values were around 24 for TGF-β and 30 for IL-10, compared to 30 for IDO in mock treated cells) (Fig 6B). In contrast, we saw significant upregulation of IDO at the RNA level in all cells treated overnight with a high dose of SEV (Fig 6C, mean 122-fold upregulation, p = 0.026 by paired T test), and a dose-dependent decrease in IDO expression with exposure to fewer SEV (Fig 6C). Even a shorter 6-hour exposure of DC to SEV resulted in a significant increase in IDO expression (mean 14.3-fold upregulation, p = 0.032, Fig 6C). We also looked for IDO protein by intracellular staining in SEV-treated DC. Exposure to SEV significantly and in a dose-responsive manner upregulated IDO expression at the protein level (Fig 6D and 6E). The SEV-treated DC had a mean of 16.8% of cells expressing high levels of IDO, versus 2.8% of mock treated cells (p = 0.002 by paired T test). Three individuals hardly increased IDO protein, despite high upregulation of IDO mRNA (compare mRNA expression and protein expression in Fig 6C and 6D). These results indicate that SEV impair APC immune function by upregulating IDO, with the interesting facet that in some individuals IDO mRNA and protein induction correlate whereas in others they appear decoupled.

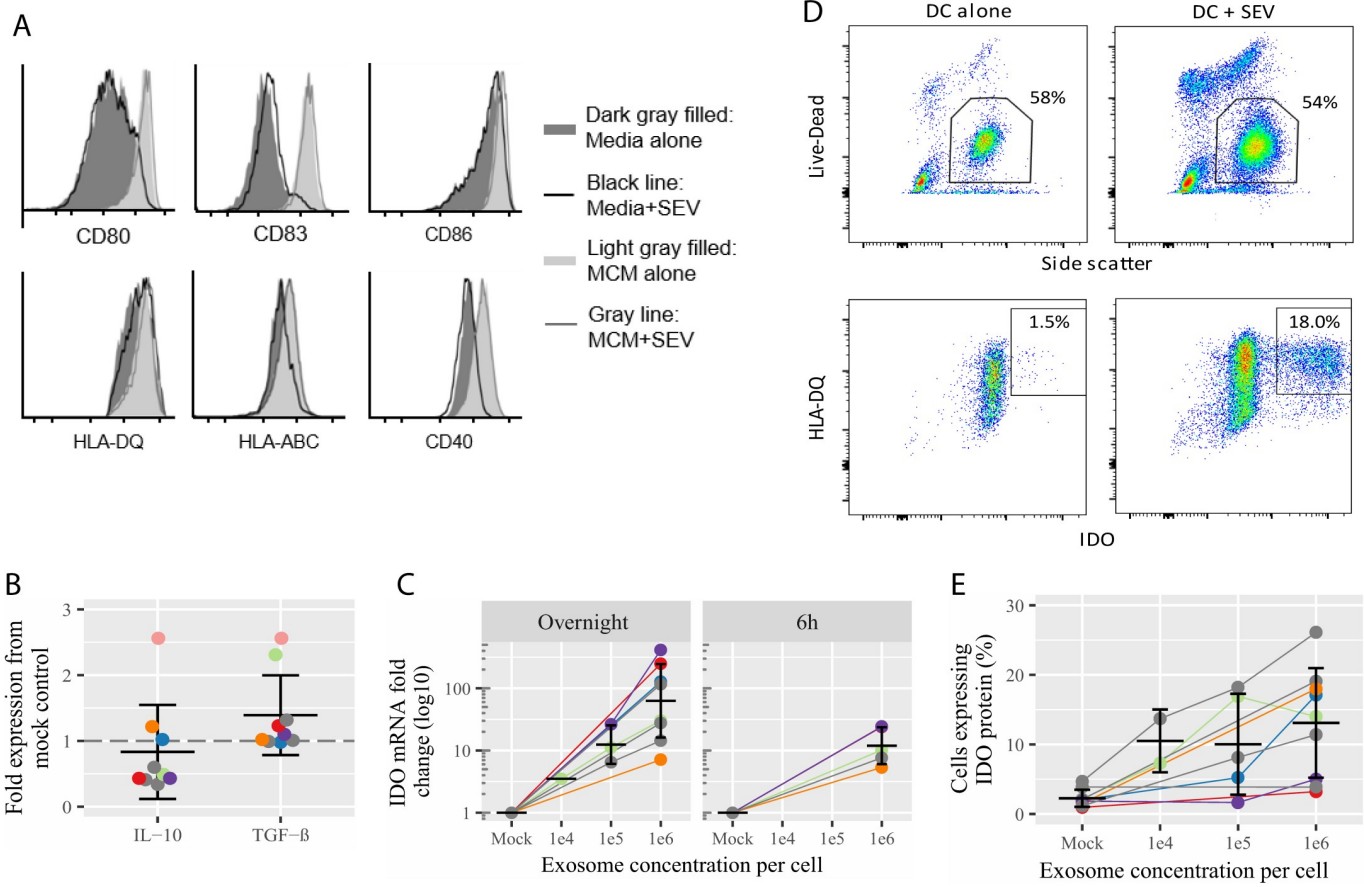

**Fig 6. MHC and co-stimulatory marker, and IDO expression, in SEV-treated dendritic cells (DC).** (A) Monocyte-derived DC were left alone or treated with a maturation cocktail (monocyte-conditioned medium, MCM), with or without $10^5$ SEV per cell. Cells were stained for surface expression of the indicated markers and analyzed by flow cytometry. One representative donor of n = 3 tested is presented. (B) DC treated overnight with $10^6$ SEV per cell were analyzed for IL-10 and TGF-β mRNA expression by qRT-PCR. Fold expression relative to mock treated DC is presented. Gray color indicates blood donors not used in previous experiments. The other colors match donors in previous experiments. (C) DC treated overnight or for 6 hours with $10^4$, $10^5$ or $10^6$ SEV per cell were analyzed for IDO mRNA expression by qRT-PCR. Fold change of IDO expression in SEV-treated cells compared to mock-treated cells was calculated by the delta-delta Ct method. (D) Gating scheme for calculating the percent of cells expressing high levels of IDO protein in one representative donor. Cells treated overnight as in C were analyzed for intracellular IDO protein expression by flow cytometry. Cells were gated on singlets (not shown), live high side scatter cells, then IDO expression in HLA-DQ$^+$ cells. Percent of cells plotted in (E) are based on the IDO gates shown here. (E) Plot of the percent of cells from 8 different donors expressing high IDO after treatment with $10^4$, $10^5$, or $10^6$ SEV per cell overnight. High IDO is defined as in D.

## Discussion

In this study, we report that extracellular vesicles from semen impair antigen-specific T cell responses, as evidenced by decreased cytokine production and cytotoxicity of CD8+ T cells, primarily by altering the function of antigen-presenting cells. As SEV share many properties with sexually transmitted viruses like HIV and Zika virus (similar size and biological composition), SEV are bound to access mucosal APCs simultaneously with penetrating viruses. Specific cellular receptors for either viruses or SEV may alter exactly which cells uptake these particles, but they will be present in the same immediate regions of the recipient mucosa. Inducing tolerance via effects on APC function makes evolutionary sense, because anti-sperm immunity, which can prevent successful conception and therefore must be restricted, is analogous to transplant rejection and requires inhibition of allogeneic adaptive, not innate, immunity. A potential down-side, however, is the compromised adaptive immune response to viral antigens we observed. It is intriguing to speculate that the reported innate antiviral effect of SEV on

HIV and Zika virus has arisen to counterbalance SEV's tolerizing effect on adaptive immunity [22, 49, 50].

The observation that SEV readily bound to and entered APCs, but not T cells (Fig 1), suggested that SEV act through APCs to reduce memory T cell responses. It could be possible that a small fraction of SEV-bound T cells affects the whole T cell population, for example by altered cytokine production. However, our subsequent experiments (Figs 2–4) demonstrate that SEV do act directly on APCs to inhibit CD8+ T cell function. First, inhibition of cytokine production was more robust when T cell activation required interaction with an APC (32% reduction on average for SEB-stimulated cultures, compared to 16% reduction for PMA/ionomycin-stimulated cultures, Fig 3A & 3B), implying that APC-T cell interactions were affected by SEV. Second, exposing DC alone to SEV was sufficient to inhibit cytokine production and degranulation by CD8+ T cells (Figs 3C & 4C). Third, exposing CMV-loaded target DC to SEV impaired the ability of CMV-specific CD8+ T cells to kill the targets (Fig 4D & 4E), indicating that SEV modify how APCs are recognized by T cells. Fourth, CD8+ T cell responses were more strongly compromised by SEV when the stimulus was a protein rather than a peptide antigen (Fig 2B). This suggests that SEV interfere with APC cross-presentation, which is required to load protein-derived but not pre-processed peptide antigens onto MHC class I molecules [51–54]. These four lines of evidence show that SEV suppress adaptive CD8+ T cell immunity primarily by acting through APCs.

Inhibition of CD4+ T cell function appears to occur via a distinct mechanism, likely at the level of interaction between APCs and T cells, because exposing DC or T cells separately to SEV did not cause the same level of impaired cytokine production as seen in mixed PBMC cultures (Fig 3C). Inhibition of cytokine production in CD4+ T cells stimulated with SEB was greater than in cells stimulated with PMA/ionomycin-stimulated cultures (32.38% for SEB vs 16.16% for PMA/ionomycin, Fig 3A & 3B), again implying that APC-T cell interactions were affected by SEV. Since SEV did also inhibit CD4+ (and CD8+) T cell responses to direct stimulation by PMA/ionomycin, it appears they directly affect T cells in certain contexts. In these experiments we did not test whether cross-presentation of cytosolic antigens on MHC II to CD4+ T cells is also impaired by SEV. Interestingly, we observed quite a bit of variation in the susceptibility of individual recipients to immunosuppression mediated by SEV (Fig 5). This suggests that certain recipient/SEV combinations may be prone to effective inhibitions where others are not. We do not yet know the mechanism of this effect. Dissecting the multifaceted effects of SEV on T cell responses and the molecules responsible are important areas for further exploration.

Of note, the antigens we used for testing antiviral T cell responses were peptides and lysates from EBV and CMV. While these viruses can be sexually transmitted [55, 56], they are also often acquired via other routes as well. We did this for practical reasons, because these infections, and the resulting T cell responses, are common, which allowed us to identify enough volunteers where preliminary testing indicated the presence of T cells reactive to these pathogens. We did not formally prove that our findings extend to immunity against HIV, Zika virus and other sexually transmitted viruses, but contend that this is very likely. Supporting this conclusion is the finding that the inhibitory effect of SEV extended beyond EBV and CMV to staphylococcal enterotoxin B stimulation.

Prostaglandins are present in semen and have immunosuppressive effects, including causing expression of high levels of immunosuppressive cytokines [2, 57]. If prostaglandins were present at high levels in SEV, they could explain the immunosuppressive effects we observed. However, the prostaglandin concentration in our SEV preparations was low, measured by ELISA at about 55 ng/mL PGE-2, as compared to 5852 ng/mL in SEV-depleted seminal plasma. At this concentration, our assays with SEV were conducted with a maximum of 5.5

ng/ml of PGE2, or about 16 nM, a dose unlikely to have a measurable functional effect [58, 59]. The much higher PGE-2 concentrations in seminal fluid, however, do highlight that considering the role of the soluble fraction will be important to fully understand the immunological effect of semen on the recipient mucosa. Different mechanisms of immunosuppression induced by SEV and soluble factors may add or even synergize toward powerful transient immunosuppression upon semen exposure.

In monocyte-derived DC, the presence of SEV during DC maturation did not change the expression of surface-exposed MHC class I or II, CD40, CD80, CD83, or CD86 (Fig 6A). These data fit the notion that SEV impair internal processes such as antigen cross-presentation or expression of tolerogenic molecules such as IDO, rather than APC activation generally. Expression of IDO is a key feature of tolerogenic DC and its activity is well-recognized in peripheral tolerance and immune regulation. IDO is an enzyme that catabolizes the essential amino acid tryptophan. Both starvation by depletion of trypotophan, and direct effects by metabolites of this process, such as kynurenine, can suppress effector T cell function and promote the generation of T regulatory cells [44]. We found that SEV very strongly induced the transcriptional upregulation of IDO in all tested DC and increased the fraction of cells expressing high levels of IDO protein in 5 of 8 individuals tested (Fig 6C). Other reports support the concept that in DC, IDO is subject to both transcriptional and translational regulation, with production of functional IDO requiring both primary stimulation and a maturation signal [60]. We also looked for regulation of expression of IL-10 and TGF-β in SEV-treated APCs. These cytokines can act on APCs in an autocrine manner to induce IDO expression [45, 46], and/or themselves suppress signaling and cytokine production in T cells [48]. Interestingly, neither of these cytokines were regulated in blood-derived DCs exposed to SEV. However, the activity of IDO alone is sufficient to impair cytokine production in co-cultured T cells [44, 60, 61]. Thus, it is likely that the downstream suppression of memory T cell function we observe is due at least in part to IDO expression.

EV causing APCs to shift toward immunosuppressive phenotypes, functionally achieved by EV-delivered regulatory RNAs including Y RNA fragments and miRNAs, has been shown to occur in cancer biology, resulting in decreased host immunity to tumors [62, 63]. Though we do not yet know the mechanism by which SEV induce IDO expression and suppress antigen presentation by APCs, exposed phosphatidylserine (PS) on SEV could contribute to their overall tolerogenic effect. Externalized PS functions as an immunosuppressive signal to prevent immune activation during cleanup of apoptotic cells [37]. This suppressive effect has been hijacked by numerous pathogens which express PS to facilitate infection and establish latency [64, 65]. It has also been shown that EV from ovarian cancer cells arrest T cell signaling via a PS-dependent mechanism [39]. Our data show that blocking PS on SEV with annexin substantially reduces SEV binding to and uptake by APCs (Fig 1D). This indicates that APCs likely do use PS receptors for recognition and uptake of SEV, a finding supported in other studies [66–68], though we cannot rule out that annexin binding sterically blocked molecules other than PS on SEV important for cell interactions. There are over a dozen PS receptors, so the exact mechanism by which immune suppression is facilitated by PS is likely overlapping and highly dependent on cell type and activation state [64]. With the evolutionary pressure to maintain tolerance to support conception, it is easy to envision SEV taking advantage of this established immunosuppressive pathway.

APCs are essential to the initiation and amplification of immune responses, either positively or in a regulatory direction. This study reports that one component in semen, SEV, induces a tolerogenic state in APCs. Other reports show that unfractionated seminal plasma can stimulate the expression of cytokines and growth factors in vaginal epithelial cells [5, 69–72], suggesting a pro-inflammatory role of semen. This may represent a multi-faceted strategy to

defend against pathogens, by putting epithelial cells on alert and recruiting innate immune cells, while limiting APC responses to semen to diminish any potential long-term memory immune responses to semen or paternal antigens, which would be detrimental. The recruitment of T cells following intercourse may also increase the pool of cells available to interact with SEV-induced regulatory APCs to become regulatory T cells, poised to promote successful conception and pregnancy[7].

In summary, we showed that the presence of SEV during the activation of T cells suppresses their subsequent responses including cytokine production and cytotoxic function. In accordance with the observation that SEV primarily enter APCs but not T cells, the impact of SEV on CD8+ T cell function was mediated largely through APCs, likely via inducing IDO expression and impairing cross-presentation. Continuing to investigate the mechanisms of how semen alters the immune responses of recipient cells in the genital mucosa could contribute to our understanding of fertility as well as to designing vaccines against STIs.

## Supporting information

**S1 Fig. Treatment of cells with e5 SEV for 16 hours does not affect cellular viability.** Five recipients were tested, error bars indicate standard deviation for technical replicates (A) Monocyte-derived DCs were mock treated or treated with $10^5$ SEV overnight. Cells were stained with live-dead viability stain, fixed, and analyzed by flow cytometry. Cells are gated on high SSC DCs and the percent of live (unstained) cells are plotted on the y-axis. (B) Negatively selected T cells were mock treated or treated with $10^5$ SEV overnight. Cells were stained with live-dead viability stain, fixed, stained with an anti-CD3 antibody and analyzed by flow cytometry. The percent of live CD3+ cells is plotted.
(PPTX)

**S1 File. Intracellular cytokine data for all cytokines.** Data for all experiments measuring cytokines from peptide or protein stimulated cells.
(XLSX)

**S2 File. Cytokine data for SEB and PMA/Ionomycin stimulated cells.** Data for all experiments measuring cytokines from cells stimulated with SEB or PMA/Ionomycin.
(XLSX)

**S3 File. Data for all figures not included in S1 File or S2 File.** Data used to generate Figs 1A, 1B, 1C, 1D, 2B, 2C, 3B, 3C, 3D, 4A, 4B, 4C, 4E, 4G, 5A, 6B, 6C and 6E.
(XLSX)

## Acknowledgments

The authors would like to thank the semen and blood donors who contributed to this study. M. Juliana McElrath is head of the clinic where samples were collected. John Amory, Consuelo Pete, Julie Czartoski, Janine Maenza, Christine Galloway and Gina Braun assisted with collecting samples and with donor recruitment.

## Author Contributions

**Conceptualization:** Lucia Vojtech, Mengying Zhang, Martin Prlic, Elizabeth Nance, Florian Hladik.

**Data curation:** Lucia Vojtech, Claire Levy, Ruofan Wang.

**Formal analysis:** Lucia Vojtech, Sean M. Hughes, Martin Prlic, Elizabeth Nance, Florian Hladik.

**Funding acquisition:** Lucia Vojtech, Martin Prlic, Elizabeth Nance, Florian Hladik.

**Investigation:** Mengying Zhang, Veronica Davé, Claire Levy, Ruofan Wang, Fernanda Calienes.

**Methodology:** Lucia Vojtech, Mengying Zhang, Veronica Davé, Sean M. Hughes.

**Project administration:** Florian Hladik.

**Supervision:** Lucia Vojtech, Martin Prlic, Elizabeth Nance, Florian Hladik.

**Validation:** Fernanda Calienes.

**Visualization:** Sean M. Hughes.

**Writing – original draft:** Lucia Vojtech.

**Writing – review & editing:** Lucia Vojtech, Mengying Zhang, Veronica Davé, Sean M. Hughes, Ruofan Wang, Martin Prlic, Elizabeth Nance, Florian Hladik.

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
