## [Decision Letter · Decision Letter 0]

2 Aug 2019

PONE-D-19-16692

Extracellular vesicles in human semen modulate antigen-presenting cell function and decrease downstream antiviral T cell responses

PLOS ONE

Dear Dr Vojtech,

Thank you for submitting your manuscript to PLOS ONE. After careful consideration, we feel that it has merit but does not fully meet PLOS ONE’s publication criteria as it currently stands. Therefore, we invite you to submit a revised version of the manuscript that addresses the points raised during the review process.

Please address reviewer comments.

EVISION_DUE%. To enhance the reproducibility of your results, we recommend that if applicable you deposit your laboratory protocols in protocols.io, where a protocol can be assigned its own identifier (DOI) such that it can be cited independently in the future. For instructions see: http://journals.plos.org/plosone/s/submission-guidelines#loc-laboratory-protocols

We look forward to receiving your revised manuscript.

Kind regards,

Geetha P. Bansal, Ph.D.

Academic Editor

PLOS ONE

Journal Requirements:

Reviewers' comments:

Reviewer's Responses to Questions

**Comments to the Author**

1. Is the manuscript technically sound, and do the data support the conclusions?

Reviewer #1: Partly

2. Has the statistical analysis been performed appropriately and rigorously? 

Reviewer #1: Yes

3. Have the authors made all data underlying the findings in their manuscript fully available?

Reviewer #1: Yes

4. Is the manuscript presented in an intelligible fashion and written in standard English?

Reviewer #1: Yes

5. Review Comments to the Author

Reviewer #1: In this manuscript, Vojtech et al. describe the uptake of semen extracellular vesicles (SEV) in APC cells, and the effect of SEV on T cell responses. The authors show that SEV alter cytokine production and activation of T cells. Although extracellular vesicles are widely studied, semen derived extracellular vesicles and their role on immune responses is understudied. The authors here contribute to the immunomodulatory function of SEV. Although the data support the argument that the SEV effects on T cell responses are primarily due to SEV effects on APC cell functions rather than a direct effect on T cells, the data show some direct effect of SEV on T cells (Fig. 3B). In fact, the PMA+ionomycin data appear to show a dichotomous relationship between SEV and T cell activation. Nevertheless, the data provide evidence that SEV effect APC cell functions that alter T cell responses quite convincingly. Further experiments on the direct effect of SEV on T cell function, or revision of the conclusion to point out that both APC and T cell function are regulated by SEV would strengthen the conclusions of the manuscript.

Major comments:

1. The authors used 105 SEV particles per cell---what does this roughly equate to as protein concentration? Do the authors have data demonstrating the effect of this number of particles on cellular viability? High particle concentrations may have cytotoxic effects on lymphocytes and DC cells which may contribute to cell function. This is especially important since the authors are often comparing SEV treatment to no treatment.

2. Lines 349-401/Fig.3— The authors show that only a small fraction of T cells take up DiI-labeled SEV. It is possible that SEV could regulate T cells when only a portion of the cells are altered (via cytokines, etc.).

3. The conclusion that SEV-mediated immunosuppression occurs upstream of protein kinase C activation (line 345-346) is not strongly supported by the data. Fig. 3A shows significant inhibition of cytokine production by SEV in both SEB and PMA/ionomycin activated cells. Without dose response data it is difficult to conclude that differences in T cell responses between the two stimuli are APC effects and not concentration or toxicity effects. In addition, the data may be strengthened if the authors included data where these stimulating molecules were used after T cell selection instead of total PBMC. Cytokine production in +/- SEB and PMA-ionomycin activated T cells +/- SEV may help address the role of APC in T cell activation. The authors should rephrase this conclusion as the data here do not clearly show that APC predominately inhibit T cell responses (lines 357-358).

4. In addition, and as noted above, the PMA+ionomycin data appear to show a dichotomous relationship between SEV and T cell activation. It would strengthen the manuscript to comment on the possible reasons for this observation.

5. A potential concern is the method of T cell purification. Was this positive or negative selection? If positive selection was used (with anti-CD3), TCR stimulation would influence the results considerably. This needs to be clarified.

6. Fig. 3C is informative in that it addresses T and DC cell populations. Adding a title to the x axis would help clarify that this axis refers to the cell population that was exposed to SEV before co-culture.

7. Why does SEV not effect CD8+ T cell cytokine levels when APCs in total PBMC are treated with CMV peptide, but SEV does affect CD8+ T cell levels when DCs are treated with CMV peptide? It would strengthen the paper if the authors’ provided their thoughts on this result.

8. The authors argue that SEV induce immune tolerance and hypothesize that this tolerance may alter transmission of sexually transmitted viruses. It would strengthen the authors argument to include APC/T cell function during a virus infection. DCs are highly susceptible to Zika virus infection, and SEV were previously shown to inhibit Zika virus infectivity (in Vero cells). The potential impact of the authors findings would be strengthened if they determined whether SEV still induces immune tolerance in Zika (or another virus) infected cells and whether SEV alters infection levels in those cells. This experiment seems important since the authors observed differences in cell responses to virus derived lysates and virus derived peptides (Fig. 2B).

Minor comments:

1. The authors argue that the similarity in size of exosomes and viruses (Fig. 1A) indicates that their mucosal penetration routes are likely to overlap (line 64-65 and 268). This conclusion may well occur, but the use of virus-specific cellular receptors may lead to greatly different modes of mucosal penetration. This should be noted. Further, it is unclear if the viruses included in Fig. 1A were measured by NTA or graphed based on literature descriptions.

2. Fig 1B—it may be important to differentiate between bound SEV and internalized SEV (trypsin cleavage of bound SEV) on the different cell types since the authors argue in 1D that DCs and monocytes actively internalize SEV (lines 302-303).

3. Line 294—reference to Fig. 1C should be 1D. Fig. 1D--What are the methods for the experiment with Langerhans cells from vaginal mucosal tissue? Without an understanding of the vaginal mucosal tissue methods the authors should not state line 308---“…and the evidence here that SEV penetrate the vaginal mucosa…”.

4. The authors show SEV uptake in APCs derived from blood, but do not show evidence that SEV traverse the vaginal mucosa. This statement should be clarified (as a hypothesis).

5. It would be more informative to include quantitation in 1D from multiple fields.

6. Figure 2A—If 10 individual donors were assayed, why do the panels vary in the number of donors presented?

7. Define which panels correspond to CD4+ and which correspond to CD8+ cells either in the figure or legend.

8. It would be more informative if the authors presented production of IFNy, IL-2, and TNFa independently (perhaps as supplemental) in addition to the sum shown in 2A (also for Fig. 3).

9. The authors’ should include statistical evaluation of the CMV lysate in Figure 3 (line 317-318).

10. The graphs would be visually improved by indication of the mean as either a line or bar when comparing multiple donors (for example 2A,3A) as the authors did for other panels (2B,3B-C).

11. Fig. 5A—the different populations are difficult to distinguish, a more typical legend or more distinct representation for the different groups would be helpful. Fig. 5B—Which concentration of SEV is graphed? Fig. 5D—Panels on the left visually appear to have fewer cells than the panels on the right. Do you see the same effect if the same number of cells are analyzed, or do you see less IDO because you’re analyzing fewer cells? Fig. 5E—Why are there 2 points for 1e4 and 9 points for 1e6 if 8 donors were assayed?

12. Clear reference to negative controls is needed in some figures. For example, the methods state that stimulation experiments were completed with negative control wells (lines 147-148), but it’s unclear if unstimulated controls were used for data normalization/analyses. This clarification may be important for understanding the stimulation experiments. Fig. 2A EBV peptide (left panel) shows less than 1% of cells responding, an understanding of cell response to negative control is needed to evaluate whether these cells responded to EBV peptide and if conclusions can be interpreted from this result. Instead of “none”, a better SEV control would be to use at least an equal volume of SEV resuspension media.

6. PLOS authors have the option to publish the peer review history of their article (what does this mean?). If published, this will include your full peer review and any attached files.

Reviewer #1: No

---

## [Author Response · Author response to Decision Letter 0]

15 Aug 2019

We thank the reviewer for their comments and have incorporated most of the requested changes and clarification, as outlined below and in the attached revised manuscript. Due to changes, line numbers have changed since the original submission, and the line numbers in our responses correspond to the revised text with tracked changes. 

Reviewer #1: In this manuscript, Vojtech et al. describe the uptake of semen extracellular vesicles (SEV) in APC cells, and the effect of SEV on T cell responses. The authors show that SEV alter cytokine production and activation of T cells. Although extracellular vesicles are widely studied, semen derived extracellular vesicles and their role on immune responses is understudied. The authors here contribute to the immunomodulatory function of SEV. Although the data support the argument that the SEV effects on T cell responses are primarily due to SEV effects on APC cell functions rather than a direct effect on T cells, the data show some direct effect of SEV on T cells (Fig. 3B). In fact, the PMA+ionomycin data appear to show a dichotomous relationship between SEV and T cell activation. Nevertheless, the data provide evidence that SEV effect APC cell functions that alter T cell responses quite convincingly. Further experiments on the direct effect of SEV on T cell function, or revision of the conclusion to point out that both APC and T cell function are regulated by SEV would strengthen the conclusions of the manuscript.

We have revised the interpretation of the data and the conclusions to take this concern into consideration. Further details are provided below, edits include lines 374-375 and lines 572-576.

Major comments:

1. The authors used 105 SEV particles per cell---what does this roughly equate to as protein concentration? 

We measured the total protein concentration in pools of SEV we used for these experiments. We found a mean of 10.61 mg of protein in 4.3035 x 1012 SEV. So 0.002465 pg per SEV x 105 SEV = approximately 332 pg of protein in 105 particles. 

Do the authors have data demonstrating the effect of this number of particles on cellular viability? High particle concentrations may have cytotoxic effects on lymphocytes and DC cells which may contribute to cell function. This is especially important since the authors are often comparing SEV treatment to no treatment.

We agree this is an important consideration. We are now including as a supplemental figure (and also attached below) data showing there is no significant effect of 105 SEV treatment of either DC or T cell viability for 16 hours. 

2. Lines 349-401/Fig.3— The authors show that only a small fraction of T cells take up DiI-labeled SEV. It is possible that SEV could regulate T cells when only a portion of the cells are altered (via cytokines, etc.).

Because incubating T cells alone with SEV did not alter cytokine production in stimulated T cells, we do not believe this is the case. However, we have changed the discussion to highlight this possibility (lines 572-576). 

3. The conclusion that SEV-mediated immunosuppression occurs upstream of protein kinase C activation (line 345-346) is not strongly supported by the data. Fig. 3A shows significant inhibition of cytokine production by SEV in both SEB and PMA/ionomycin activated cells. Without dose response data it is difficult to conclude that differences in T cell responses between the two stimuli are APC effects and not concentration or toxicity effects. In addition, the data may be strengthened if the authors included data where these stimulating molecules were used after T cell selection instead of total PBMC. Cytokine production in +/- SEB and PMA-ionomycin activated T cells +/- SEV may help address the role of APC in T cell activation. The authors should rephrase this conclusion as the data here do not clearly show that APC predominately inhibit T cell responses (lines 357-358).

Since SEV do inhibit cytokine production in PMA/ionomycin stimulated cells, the inhibition is upstream of PKC. But we do appreciate the point that the mechanism of T cell activation between SEB and PMA/ionomycin stimulation is different and might be concentration or time dependent, which we did not investigate. To make the conclusions less broad and to highlight the fact that the effects of SEV might not be entirely on APCs, we have changed the text of the manuscript (lines 358-375).

5. A potential concern is the method of T cell purification. Was this positive or negative selection? If positive selection was used (with anti-CD3), TCR stimulation would influence the results considerably. This needs to be clarified.

T cell selection was always negative selection, leaving the T cells untouched. We have added negative selection to line 211 in the methods to clarify

4. In addition, and as noted above, the PMA+ionomycin data appear to show a dichotomous relationship between SEV and T cell activation. It would strengthen the manuscript to comment on the possible reasons for this observation.

This comment inspired us to look more closely into the variation of susceptibility to immunosuppression by SEV between recipients. We plotted the percent inhibition for each assay, by recipient and found an interesting pattern where some recipients were consistently impaired by SEV while others were more refractory. The differences between individuals was significant by one-way ANOVA. We also found strong correlation between impairment in cytokine production and in CD107a expression/degranulation capacity. We have created a new figure and text to highlight this dichotomy, which we would like to include as Fig 5, if the reviewer and editor agree. The proposed text, figure, and legend are pasted below. We have also added text to the discussion in lines 597-600 to highlight this observation. Additionally, we indicated in the discussion that we do not yet know the mechanism of this dichotomy. 

Variability of SEV-mediated T cell inhibition between individuals

Our SEV inhibition assays showed large variability between the individuals tested. Having performed several different functional assays in each SEV recipient gave us the opportunity to assess whether some people are intrinsically more susceptible to SEV inhibition than others. Indeed, some individuals showed consistently strong impairment of their T cell responses by SEV no matter what functional assay was used (e.g., individuals H and E in Fig 5A). In contrast, one person was comparatively refractory to SEV inhibition across all assays tested (individual C in Fig 5A). Overall, the difference in impairment of CD8+ T cell responses mediated by SEV between individuals was significant by one-way ANOVA (Fig 5A) and the susceptibilities to decreased cytokine production and reduced CD107a expression were correlated (Pearson r=0.724) (Fig 5B). These results indicate that at least some individuals possess high intrinsic susceptibility to SEV-mediated inhibition of their T cell responses, whereas others may be mostly refractory. 

6. Fig. 3C is informative in that it addresses T and DC cell populations. Adding a title to the x axis would help clarify that this axis refers to the cell population that was exposed to SEV before co-culture.

We had added a title as requested

7. Why does SEV not effect CD8+ T cell cytokine levels when APCs in total PBMC are treated with CMV peptide, but SEV does affect CD8+ T cell levels when DCs are treated with CMV peptide? It would strengthen the paper if the authors’ provided their thoughts on this result. 

Treatment of PBMC with SEV at the time of exposure to CMV peptide does effect CD8+ cytokine levels, in 4 of 7 recipients, as observed in figure 2A and 2B. Because 3 of 7 recipients do not have impaired CD8+ T cell responses with SEV during CMV peptide exposure, this did not reach significance. When DCs alone are treated with SEV at the time of exposure to CMV peptide, as in 3C, 5 of 7 recipients have impaired T cell responses. The differences probably have to do with the fact that in the separate exposure experiments in Fig 3, the ratio of DC to T cells is higher than in total PBMC (in Fig 2). 

8. The authors argue that SEV induce immune tolerance and hypothesize that this tolerance may alter transmission of sexually transmitted viruses. It would strengthen the authors argument to include APC/T cell function during a virus infection. DCs are highly susceptible to Zika virus infection, and SEV were previously shown to inhibit Zika virus infectivity (in Vero cells). The potential impact of the authors findings would be strengthened if they determined whether SEV still induces immune tolerance in Zika (or another virus) infected cells and whether SEV alters infection levels in those cells. This experiment seems important since the authors observed differences in cell responses to virus derived lysates and virus derived peptides (Fig. 2B).

We have a manuscript outlining a finding that SEV strongly inhibit Zika virus infection by preventing viral binding or entry into cells nearly ready to submit for publication. The goal of this paper was to determine how SEV impact already established memory immune responses. We agree that it is intriguing to consider how SEV and components of semen might impact primary viral infections, and we plan to investigate this in future studies. 

Minor comments:

1. The authors argue that the similarity in size of exosomes and viruses (Fig. 1A) indicates that their mucosal penetration routes are likely to overlap (line 64-65 and 268). This conclusion may well occur, but the use of virus-specific cellular receptors may lead to greatly different modes of mucosal penetration. This should be noted. Further, it is unclear if the viruses included in Fig. 1A were measured by NTA or graphed based on literature descriptions.

We have added text to discussion (lines 562-564) to note this point. Viruses in 1A are graphed based on literature descriptions, we have added clarification to the figure legend. 

2. Fig 1B—it may be important to differentiate between bound SEV and internalized SEV (trypsin cleavage of bound SEV) on the different cell types since the authors argue in 1D that DCs and monocytes actively internalize SEV (lines 302-303).

We have re-phrased that section (lines 310-311) to be more clear about what we know about the difference between binding and internalization. 

3. Line 294—reference to Fig. 1C should be 1D. 

Thanks for catching this! Fixed

Fig. 1D--What are the methods for the experiment with Langerhans cells from vaginal mucosal tissue? Without an understanding of the vaginal mucosal tissue methods the authors should not state line 308---“…and the evidence here that SEV penetrate the vaginal mucosa…”.

We have added methods to lines 121-126 to explain how LC experiments were done (lines 121-126), and changed former line 308, now line 317, to reflect this concern. 

4. The authors show SEV uptake in APCs derived from blood, but do not show evidence that SEV traverse the vaginal mucosa. This statement should be clarified (as a hypothesis). 

We have re-phrased lines 317 to reflect this concern.

5. It would be more informative to include quantitation in 1D from multiple fields.

It is very difficult to obtain and stain LC-T cell conjugates from tissue and we don’t believe quantification would be very meaningful in this case. it is also possible that SEV being internalized by APC fuse their membranes with cellular membranes, thus diluting the dye and diffusing the signal. We include this data only as an interesting anecdote that we have only ever observed APCs becoming stained following co-culture with SEV, and never T cells.

6. Figure 2A—If 10 individual donors were assayed, why do the panels vary in the number of donors presented?

Only recipients where T cell responses met the criteria of cytokine production of 2 times the background levels were included as responder to each particular stimuli. Not all donors responded to each antigen. We have added text to line 327 to clarify this. 

7. Define which panels correspond to CD4+ and which correspond to CD8+ cells either in the figure or legend.

Thanks. This was an accidental oversight we have fixed now. 

8. It would be more informative if the authors presented production of IFNy, IL-2, and TNFa independently (perhaps as supplemental) in addition to the sum shown in 2A (also for Fig. 3).

We will now include two supplemental files with the raw data, including cytokine production for each individual cytokine, for the data presented in Figures 2A and 3A. We have added text to the manuscript pointing towards the availability of this data. 

9. The authors’ should include statistical evaluation of the CMV lysate in Figure 3 (line 317-318).

As indicated in the figure legend, we tested all antigens for statistical significance using Wilcoxon matched-pairs signed rank test. P values have been added to figure 2A.

10. The graphs would be visually improved by indication of the mean as either a line or bar when comparing multiple donors (for example 2A,3A) as the authors did for other panels (2B,3B-C).

We have added means to the figures as requested.

11. Fig. 5A—the different populations are difficult to distinguish, a more typical legend or more distinct representation for the different groups would be helpful. 

We have changed the figure legend as requested.

Fig. 5B—Which concentration of SEV is graphed? 

We did test multiple doses of SEV for their effect on IL-10 and TGF-B. Since we did not see a significant effect on any transcript only the 106 SEV dose is plotted. We have clarified this in the figure legend. 

Fig. 5D—Panels on the left visually appear to have fewer cells than the panels on the right. Do you see the same effect if the same number of cells are analyzed, or do you see less IDO because you’re analyzing fewer cells?

Analysis of IDO expression is based on the percent of live, high side scatter, HLA-DQ high cells. The denominator shifts if cell numbers are not exactly the same. If we limit data analysis to a fixed number of cells the observation remains the same. 

Fig. 5E—Why are there 2 points for 1e4 and 9 points for 1e6 if 8 donors were assayed?

The yield of DCs after monocyte selection and culture varies between individuals. For most recipients we did not have enough cells to analyze all SEV doses, so the 1e4 dose was done only in a subset of donors. 

12. Clear reference to negative controls is needed in some figures. For example, the methods state that stimulation experiments were completed with negative control wells (lines 147-148), but it’s unclear if unstimulated controls were used for data normalization/analyses. This clarification may be important for understanding the stimulation experiments. Fig. 2A EBV peptide (left panel) shows less than 1% of cells responding, an understanding of cell response to negative control is needed to evaluate whether these cells responded to EBV peptide and if conclusions can be interpreted from this result. Instead of “none”, a better SEV control would be to use at least an equal volume of SEV resuspension media.

All negative unstimulated controls contained the exact same volumes and inputs as stimulated conditions. For example unstimulated PBMC got equal volumes of DMSO, BFA, and co-stimulatory antibodies. For experiments without SEV, all wells got equal volumes of PBS, which is the SEV washing and resuspension media. We have clarified this in the methods section (lines 155-157).

---

## [Decision Letter · Decision Letter 1]

2 Oct 2019

Extracellular vesicles in human semen modulate antigen-presenting cell function and decrease downstream antiviral T cell responses

PONE-D-19-16692R1

Dear Dr. Vojtech,

We are pleased to inform you that your manuscript has been judged scientifically suitable for publication and will be formally accepted for publication once it complies with all outstanding technical requirements.

With kind regards,

Geetha P. Bansal, Ph.D.

Academic Editor

PLOS ONE

Additional Editor Comments (optional):

Reviewers' comments:

Reviewer's Responses to Questions

**Comments to the Author**

1. If the authors have adequately addressed your comments raised in a previous round of review and you feel that this manuscript is now acceptable for publication, you may indicate that here to bypass the “Comments to the Author” section, enter your conflict of interest statement in the “Confidential to Editor” section, and submit your "Accept" recommendation.

Reviewer #1: All comments have been addressed

2. Is the manuscript technically sound, and do the data support the conclusions?

Reviewer #1: Yes

3. Has the statistical analysis been performed appropriately and rigorously? 

Reviewer #1: Yes

4. Have the authors made all data underlying the findings in their manuscript fully available?

Reviewer #1: Yes

5. Is the manuscript presented in an intelligible fashion and written in standard English?

Reviewer #1: Yes

6. Review Comments to the Author

Reviewer #1: (No Response)

7. PLOS authors have the option to publish the peer review history of their article (what does this mean?). If published, this will include your full peer review and any attached files.

Reviewer #1: No

---

## [Editor Report · Acceptance letter]

10 Oct 2019

PONE-D-19-16692R1 

Extracellular vesicles in human semen modulate antigen-presenting cell function and decrease downstream antiviral T cell responses 

Dear Dr. Vojtech:

I am pleased to inform you that your manuscript has been deemed suitable for publication in PLOS ONE. Congratulations! Your manuscript is now with our production department. 

With kind regards,

on behalf of

Dr. Geetha P. Bansal 

Academic Editor

PLOS ONE